# RefConv: Re-parameterized Refocusing Convolution for Powerful ConvNets

## Abstract

We propose Re-parameterized Refocusing Convolution (RefConv) as a replacement for regular convolutional layers, which is a plug-and-play module to improve the performance without any inference costs. Specifically, given a pre-trained model, RefConv applies a trainable Refocusing Transformation to the basis kernels inherited from the pre-trained model to establish connections among the parameters. For example, a depth-wise RefConv can relate the parameters of a specific channel of convolution kernel to the parameters of the other kernel, i.e., make them refocus on the other parts of the model they have never attended to, rather than focus on the input features only. From another perspective, RefConv augments the priors of existing model structures by utilizing the representations encoded in the pre-trained parameters as the priors and refocusing on them to learn novel representations, thus further enhancing the representational capacity of the pre-trained model. Experimental results validated that RefConv can improve multiple CNN-based models by a clear margin on image classification (up to 1.47% higher top-1 accuracy on ImageNet), object detection and semantic segmentation without introducing any extra inference costs or altering the original model structure. Further studies demonstrated that RefConv can reduce the redundancy of channels and smooth the loss landscape, which explains its effectiveness.

## 1 Introduction

Convolutional Neural Networks (CNNs) have indeed been the dominant tool for a wide range of computer vision tasks. One of the mainstream approaches to improving the performance of CNNs is to elaborately design the model structures, including macro model architectures (He et al., 2016; Huang et al., 2017a; Liu et al., 2022) and micro plug-and-play components (Hu et al., 2018; Woo et al., 2018; Li et al., 2019b). The success of CNN can be partly attributed to the locality of operations. For the spatial dimensions, a typical example is the sliding-window mechanism of convolution which utilizes the local priors of images. For the channel dimension, a depth-wise convolution (referred to as *DW conv* for brevity) operates on each input channel with an independent 2D convolution kernel, significantly reducing the parameters and computations, compared to a regular *dense conv* (which means each output channel attends to every input channel, i.e., the number of groups is 1).

In this paper, we propose to improve the performance of CNNs from another perspective - augmenting the priors of existing structures. For example, a DW conv can be regarded as a concatenation of multiple mutually independent 2D conv kernels (referred to as *kernel channels*), and the only input to a specific kernel channel is its corresponding channel of the feature map (referred to as *feature channel*), which may limit the model's representational capacity. We seek to add more priors without changing the model's definition or introducing any inference costs (e.g., letting the kernel channel operate with the other feature channels will make the operation no longer a DW conv), so we propose a re-parameterization technique to *augment the priors of model structures by making their parameters attend to the parameters of other structures*.

Specifically, we propose a technique named *Re-parameterized Refocusing*, which establishes connections among the parameters of existing structures. Given a pre-trained CNN, we replace its conv layers with our proposed *Re-parameterized Refocusing Convolution (RefConv)*, as shown in Fig. 1. Taking DW conv again as an example, a DW conv of a pre-trained CNN will be replaced by a RefConv which freezes its pre-trained conv kernel as the *basis weights* $W_b$ and apply a train-

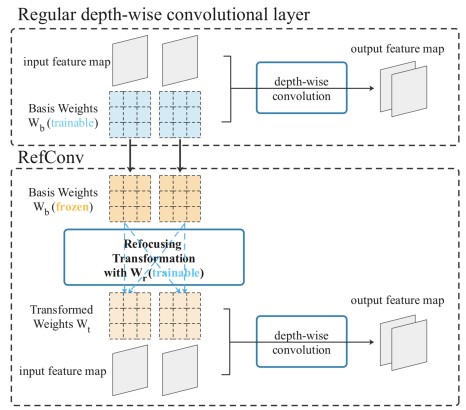
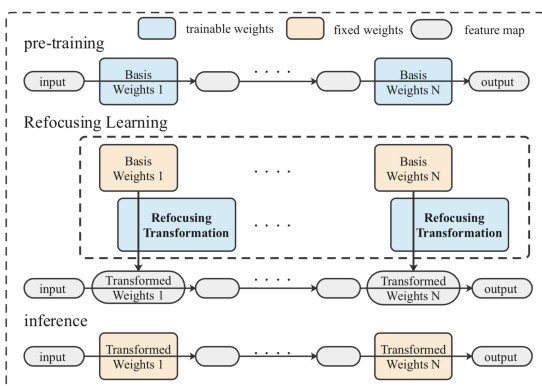

(a) Depth-wise RefConv with two input channels     (b) Two-stage pipeline to improve CNN with RefConv

Figure 1: (a) We showcase a depth-wise RefConv with two input channels, whose kernel size is $3\times3$. We apply a trainable Refocusing Transformation to the basis weights $W_b$ (which are inherited from the pre-trained model and frozen) to generate $W_t$, which then operates on the input features. A specific channel of the original model's conv kernel (i.e., a $3\times3$ matrix in this case) only attends to a single channel of the input feature map, by the definition of DW conv. In contrast, RefConv can establish connections between each channel of the conv kernel and every other channel of $W_b$ through the Refocusing Transformation. (b) We adopt a two-stage pipeline to improve CNN with RefConv. After a regular pre-training stage (which can be skipped if an off-the-shelf model is available), we construct a RefConv Model by replacing the regular conv layers by the corresponding RefConv layers which are built with the basis weights inherited from the pre-trained model. During Refocusing Learning, the basis weights are frozen and the Refocusing Transformations are learnable. Finally, we save the transformed weights $W_t$ only and use them for inference.

able operation, which is referred to as *Refocusing Transformation* $T(\cdot)$, to $W_b$ to generate a new DW conv kernel, which is referred to as *transformed weights* $W_t$. We use *refocusing weights* $W_r$ to denote the trainable parameters additionally introduced by Refocusing Transformation, so that $W_t = T(W_b, W_r)$. We then use $W_t$, instead of the original parameters, to operate on the input features. In other words, we use a different parameterization of the conv kernel. With a properly designed Refocusing Transformation, we can relate the parameters of a specific kernel channel to the parameters of the other kernel channels, i.e., make them *refocus* on the other parts of the model (rather than the input features only) to learn new representations. As the latter are trained with the other feature channels, they encode the representations condensed from the other feature channels, so that we can *indirectly* establish connections among the feature channels, which cannot be realized directly (by the definition of DW conv). After a training process (referred to as *Refocusing Learning*) of the constructed model (*RefConv Model*), we use the trained refocusing weights and the frozen basis weights to generate the final transformed weights, which are saved and used for inference only. Eventually, the resultant model (*Re-parameterized RefConv Model*) will deliver higher performance with identical inference costs to the original model. In addition, since the Refocusing Transformation in RefConv is conducted on the basis weights instead of the batches of training examples, the Refocusing Learning process is computational efficiency and memory saving.

Except for DW conv, RefConv can easily generalize to other forms such as group-wise and dense conv. As a generic design element, RefConv can be applied to any off-the-shelf CNN models with different structures. Our experimental results show that RefConv can improve the performance of multiple ConvNets on image classification, object detection and semantic segmentation by a clear margin. For example, RefConv improves MobileNetv3 (Howard et al., 2019) and ShuffleNetv2 (Ma et al., 2018) by up to 1.47% and 1.26% higher top-1 accuracy on ImageNet. To be emphasized, such performance improvements are realized with no extra inference costs or alterations to the original model structure. We further seek to explain the effectiveness of RefConv and discover that RefConv can enlarge the KL divergence between the pairs of kernel channels, which validates that RefConv can reduce the channel similarity and redundancy (Zhou et al., 2019; Wang & Stella, 2021) through attending to other channels. This enables RefConv to learn more diversified representations and enhance the model's representational capacity. In addition, it is observed that the model with RefConv has a smoother loss landscape, suggesting a better generalization ability (Li et al., 2018).

Our contributions are summarized as follows.

- We propose Re-parameterized Refocusing, which augments the priors to existing structures by establishing connections to the learned kernels. Consequently, the re-parameterized kernels can learn more diverse representations, thus further improving the representational capacity of the trained CNNs.

- We propose RefConv to replace the original conv layers and experimentally validate that RefConv can improve the performance of various backbone models on ImageNet by a clear margin without extra inference costs or altering model structure. Moreover, RefConv can also improve the ConvNets on object detection and semantic segmentation.

- We demonstrate that RefConv can reduce the channel redundancy and smooth the loss landscape, which explains the effectiveness.

## 2 RELATED WORK

### 2.1 STRUCTURE DESIGNS FOR BETTER PERFORMANCE

The designs of CNN structures for better performance include specific macro architecture and generic micro components. Representatives of macro architectures include VGGNet (Simonyan & Zisserman, 2014), ResNet (He et al., 2016), etc. (Huang et al., 2017a; Liu et al., 2022; Howard et al., 2017; Sandler et al., 2018; Howard et al., 2019) Micro components, such as SE block (Hu et al., 2018), CBAM block (Woo et al., 2018), etc. (Li et al., 2019b; Chen et al., 2019; Zhang, 2019), are usually architecture-agnostic (Ding et al., 2019), which can be incorporated into various models and bring generic benefits. However, all of these model designs change the predefined model structure. In contrast, RefConv focuses on the parameters of convolution kernels and intends to augment the priors of existing structures. As RefConv does not change the model structure, it is complementary to the advancements in the designs of architectures or components.

### 2.2 STRUCTURAL RE-PARAMETERIZATION

Structural Re-parameterization (Ding et al., 2019; 2021a;b;c; 2022a;b) is a representative re-parameterization methodology to parameterize a structure with the parameters transformed from another structure. Typically, it adds extra branches to the model in training to improve the performance, then equivalently simplifies the training structure into the same as the original model for inference. For example, ACNet (Ding et al., 2019) constructs two extra vertical and horizontal convolution branches in training and converts them into the original branch in inference. RepVGG (Ding et al., 2021c) constructs identity mappings parallel to the $3 \times 3$ convolution during training and converts the shortcuts into the $3 \times 3$ branch. Similar to the aforementioned designs in the structure space, Structural Re-parameterization builds extra branches to process the feature maps, which incurs considerable extra computational and memory costs during training. In contrast, the extra transformations in RefConv are applied to the basis weights only, which is computationally efficient and memory saving, compared to Structural Re-parameterization.

### 2.3 WEIGHT RE-PARAMETERIZATION METHODS

As a representative weight re-parameterization method, DiracNet (Zagoruyko & Komodakis, 2017) encodes the convolution kernel as the linear combination of the normalized kernel and the identity tensor. Weight normalization includes standard normalization (Salimans & Kingma, 2016), centered normalization (Huang et al., 2017b), and orthogonal normalization (Huang et al., 2018), which normalizes the weights in order to accelerate and stabilize training. These weight re-parameterization methods are independent from the data. Dynamic convolution (Zhang et al., 2020b; Chen et al., 2020), such as CondConv (Yang et al., 2019) and ODConv (Li et al., 2022), can be viewed as data-dependent weight re-parameterization, as it uses specifically-designed over-parameterized hyper-networks (Ma et al., 2020; Ha et al., 2016) which take data as the input and generate specific weights for the certain data. However, due to the dependency on the input data, such additional hyper-networks can not be removed in inference, thus introducing significant extra parameters and computational costs in both training and inference.

Refocusing Learning derives new weights with some meta weights (instead of the data), then utilizes the new weights for computations, so it can be categorized as a data-independent weight re-parameterization method.

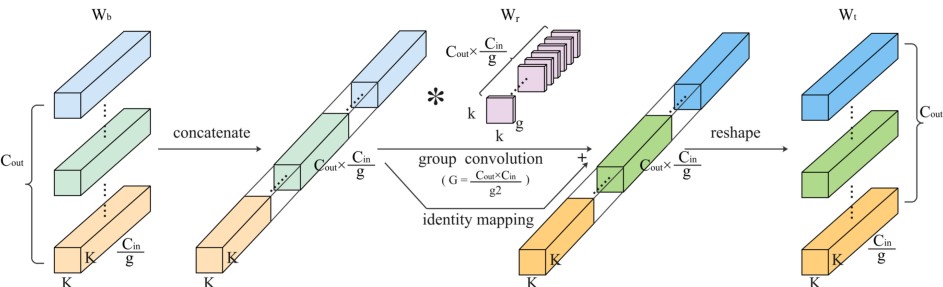

Figure 2: The general form of Refocusing Transformation.

# 3 RE-PARAMETERIZED REFOCUSING CONVOLUTION

In this section, we first elaborate on the design of a RefConv as a replacement for a regular depth-wise conv, then describe how to generalize it to group-wise or dense cases.

## 3.1 DEPTH-WISE REFCONV

Denote the number of input channels by $C_{in}$, output channels by $C_{out}$, and groups by $g$. A depth-wise convolution is configured by $C = C_{in} = C_{out} = g$. Assume the kernel size is $K$, so that the basis weights and transformed weights can be formulated as $\mathbf{W}_b, \mathbf{W}_t \in \mathbb{R}^{C \times 1 \times K \times K}$. Note that we desire not to change the model's inference structure, so that $\mathbf{W}_t$ should be of the same shape as $\mathbf{W}_b$.

We desire a proper design of the Refocusing Transformation T, which transforms the frozen $\mathbf{W}_b$ into $\mathbf{W}_t$. For a specific channel of $\mathbf{W}_t$, such a function T is expected to establish connections between it and every single channel of $\mathbf{W}_b$. In this paper, we propose to use a dense convolution as T, so that the Refocusing Transformation is parameterized by the kernel tensor of such a dense convolution $\mathbf{W}_r \in \mathbb{R}^{C \times C \times k \times k}$. We use $k = 3$ by default so that it should have padding = 1 to ensure $\mathbf{W}_t$ has the same shape as $\mathbf{W}_b$. Intuitively, such an operation can be seen as *scanning the basis weights with a $3 \times 3$ sliding window parameterized by $\mathbf{W}_r$ to extract representations to construct the desired kernel, just like we scan the feature maps with a regular conv kernel to extract patterns*. As such a convolution is dense, the inter-channel connections are established, so that each channel of $\mathbf{W}_t$ relates to all the channels of $\mathbf{W}_b$, as shown in Fig. 1(a). Just as the metaphor indicates, $\mathbf{W}_b$ can be regarded as the input "feature map" to the Refocusing Transformation, which may be designed more carefully, borrowing novel ideas from the model structure design literature. For example, we may employ non-linearity or more advanced operations, which may perform better. In this paper, we use a single convolution because it is simple, intuitive, and effective enough. Better implementations of Refocusing Transformations are scheduled as our future work.

Moreover, as inspired by residual learning (He et al., 2016), we desire the Refocusing Transformation to learn the increments over the basis weights, rather than the original mapping, just like we use the residual blocks to learn the increments over the base feature maps in ResNets. Therefore, we add a similar "identity mapping", so that

$$\mathbf{W}_t = \mathbf{W}_b * \mathbf{W}_r + \mathbf{W}_b \,, \qquad (1)$$

where $*$ denotes convolution operator.

## 3.2 GENERAL REFCONV

For a general dense or group-wise RefConv, where the basis weights and transformed weights are denoted by $\mathbf{W}_b, \mathbf{W}_t \in \mathbb{R}^{C_{out} \times \frac{C_{in}}{g} \times K \times K}$, we generalize the Refocusing Transformation designed for the depth-wise case. Just like the depth-wise case transforms the $C_{out}$ basis kernel channels into $C_{out}$ transformed kernel channels, in the general case we transform the $C_{out}$ basis kernel slices (each with $\frac{C_{in}}{g}$ channels) into $C_{out}$ transformed kernel slices. We still use convolution as the Refocusing Transformation, so that it should be configured with output channels = input channels = $C_{out} \times \frac{C_{in}}{g}$, which may be large if $\mathbf{W}_b$ is dense ($g$=1). To reduce the parameters of such a Refocusing Transformation, we make it group-wise, introducing its number of groups $G$ as a hyper-parameter, so that its number of parameters is $\frac{C_{out}^2 C_{in}^2 k^2}{g^2 G}$.

We propose a formula to determine $G$ as

$$G = \frac{C_{out}C_{in}}{g^2} . \tag{2}$$

Such a design makes the Refocusing Transformation complementary to the original convolution: a larger $g$ means a sparser original convolution, which needs more cross-channel connections established by Refocusing Transformation; according to Eq. 2, a larger $g$ results in a smaller $G$, which exactly meets such a demand.

For example, if $W_b$ is dense, we have $G = C_{out}C_{in}$ and $W_r \in \mathbb{R}^{(C_{out}C_{in}) \times 1 \times k \times k}$, which is a depth-wise convolution with $C_{out}C_{in}$ kernel channels, so it only aggregates the learned representations across the spatial dimensions but performs no cross-channel re-combinations (which are not desired since $W_b$ can operate across the feature channels by itself). On the contrary, if $W_b$ is a depth-wise kernel, we will have $G = 1$ and $W_r \in \mathbb{R}^{C_{out} \times C_{in} \times K \times K}$, which is exactly the dense convolution kernel discussed in Sec. 3.1, which fully establishes the required cross-channel connections.

We would like to note that RefConv adds only minor extra computations during training. Assume the feature map is of $B \times C_{in} \times H \times W$, the FLOPs of the original convolution will be $\frac{BHWC_{in}C_{out}K^2}{g}$, while the FLOPs of Refocusing Transformation is only $\frac{K^2 C_{out}^2 C_{in}^2 k^2}{g^2 G} = K^2 k^2 C_{in} C_{out}$, which is irrelevant to the batch size $B$. For example, assume $B = 256$, $H=W=28$, $C_{in}=C_{out}=g$=512, $K$=3 (common case of a DW layer in a regular CNN trained on ImageNet), the FLOPs is 925M for the original conv while only 21M for the Refocusing Transformation. The detailed computation, including the identity mapping and necessary reshaping operations, is depicted in Fig. 2.

### 3.3 REFOCUSING LEARNING

Refocusing Learning begins with a given pre-trained CNN, which can be obtained through a regular pre-training stage if we have no available off-the-shelf model. We construct a RefConv Model by replacing the regular conv layers with the corresponding RefConv layers. We do not replace the $1 \times 1$ conv layers because they are dense in channel and encode no spatial patterns, hence there is no need to establish cross-channel connections nor extract spatial representations from it. The RefConv layers are built with the $W_b$ inherited from the pre-trained model and $W_r$ initialized with Xavier random initialization. Moreover, $W_r$ can be initialized as zeros to make the initial model equivalent to the pre-trained model (since $W_t = W_b$ for every RefConv), which is tested in Sec. 4.4.

During Refocusing Learning, a RefConv layer computes the transformed weights $W_t = T(W_b, W_r)$, where $W_b$ is fixed and $W_r$ is learnable, and uses $W_t$ to operate on the input features. Therefore, the gradients will back-propagate through $W_t$ to $W_r$, so that $W_r$ will be updated by the optimizer just like the routines of training a regularly parameterized model.

After Refocusing Learning, we compute the final transformed weights with $W_b$ and the trained $W_r$. We save the final transformed weights only and use them as the parameters of the original CNN for inference. In this way, the inference-time model will have exactly the same structure as the original.

## 4 EXPERIMENTS

### 4.1 PERFORMANCE EVALUATION ON IMAGENET

**Dataset and models**. We first conduct abundant experiments to validate the effectiveness of Ref-Conv in enhancing the representational capacity and improving the performance of CNNs on ImageNet (Russakovsky et al., 2015), which is one of the most widely used but challenging real-world benchmark datasets for computer vision. ImageNet comprises 1.28M images for training and 50K images for validation from 1,000 classes. We experiment with multiple representative CNN architectures, covering different types of convolution layers (namely, DW conv, group-wise conv and dense conv). The tested CNNs include MobileNetv1,v2,v3 (Howard et al., 2017; Sandler et al., 2018; Howard et al., 2019), MobileNeXt (Zhou et al., 2020), HBONet (Li et al., 2019a), Efficient-Net (Tan & Le, 2019), ShuffleNetv1,v2 (Zhang et al., 2018; Ma et al., 2018), ResNet (He et al., 2016), DenseNet (Huang et al., 2017a), FasterNet (Chen et al., 2023) and ConvNeXt (Liu et al., 2022).

**Configurations**. For training the baseline models, we adopt an SGD optimizer with momentum of 0.9, batch size of 256, and weight decay of $4 \times 10^{-5}$, as the common practice (Ding et al., 2022b). We

Table 1: Results of RefConv models (RefC.) and the normally trained baselines (Base.) on ImageNet. We also report the number of training-time parameters, FLOPs and memory costs of baselines and RefConv models. To emphasize that the inference-time parameters and FLOPs of the final Re-parameterized RefConv model are identical to those of the corresponding baseline.

| Index | Top-1 Accuracy | | Params (M) | | FLOPs (G) | | Memory (G) | |
|---|---|---|---|---|---|---|---|---|
| Model | Base. | RefC. | Base. | RefC. | Base. | RefC. | Base. | RefC. |
| MobileNetv1 | 72.18% | 72.96% (**+0.82%**) | 3.22 | 28.29 | 150.76 | 150.96 | 19.83 | 20.21 |
| MobileNetv2 | 71.68% | 72.35% (**+0.67%**) | 3.56 | 44.11 | 90.37 | 90.72 | 24.21 | 24.98 |
| MobileNetv3-S | 61.95% | 63.42% (**+1.47%**) | 2.94 | 11.15 | 17.12 | 17.20 | 14.49 | 14.68 |
| MobileNetv3-L | 71.73% | 72.91% (**+1.18%**) | 5.48 | 34.06 | 61.15 | 61.41 | 24.33 | 24.85 |
| MobileNeXt | 71.57% | 72.81% (**+1.24%**) | 3.31 | 109.35 | 79.37 | 80.92 | 30.29 | 32.21 |
| HBONet | 71.61% | 72.59% (**+0.98%**) | 4.56 | 44.49 | 83.71 | 84.10 | 25.26 | 25.66 |
| EfficientNet-B0 | 75.78% | 76.74% (**+0.96%**) | 4.98 | 72.67 | 103.53 | 104.20 | 31.02 | 31.78 |
| ShuffleNetv1 | 63.17% | 64.30% (**+1.13%**) | 1.81 | 4.56 | 35.52 | 35.55 | 14.61 | 14.82 |
| ShuffleNetv2 | 67.66% | 68.92% (**+1.26%**) | 2.28 | 5.94 | 39.65 | 39.69 | 13.17 | 13.30 |
| ResNet-18 | 70.69% | 71.63% (**+0.94%**) | 11.72 | 22.74 | 472.08 | 472.20 | 15.52 | 15.76 |
| ResNet-50 | 76.16% | 76.96% (**+0.80%**) | 25.61 | 36.97 | 1063.35 | 1063.55 | 32.14 | 32.54 |
| ResNet-101 | 77.14% | 77.72% (**+0.68%**) | 44.63 | 66.01 | 2018.72 | 2018.96 | 42.97 | 43.92 |
| DenseNet-169 | 76.17% | 76.90% (**+0.73%**) | 14.18 | 17.24 | 884.81 | 884.95 | 49.10 | 49.96 |
| FasterNet-S | 78.76% | 79.91% (**+1.15%**) | 28.44 | 34.65 | 1091.28 | 1091.85 | 24.24 | 24.62 |
| ConvNeXt-T | 80.82% | 81.68% (**+0.96%**) | 28.59 | 57.37 | 1139.10 | 1139.97 | 38.53 | 38.62 |

use a learning rate schedule with a 5-epoch warmup, initial value of 0.1, and cosine annealing for 100 epochs. The data augmentation uses random cropping and horizontal flipping. The input resolution is $224 \times 224$. For Refocusing Learning, we initialize the weights of Refocusing Transformations with Xavier random initialization (Glorot & Bengio, 2010) and freeze the basis weights inherited from the corresponding pre-trained models. Refocusing Learning uses the same optimization strategy as the baselines. Besides, we make no difference to the final model architectures.

**Performance improvements**. Table 1 shows the experimental results. It can be observed that RefConv can significantly boost the performance of various baseline models with a clear margin. For example, RefConv improves the top-1 accuracy of MobileNetv3-S (DW Conv), ShuffleNetv2 (group-wise conv) and FasterNet-S (DW and dense conv) by 1.47%, 1.26% and 1.15%, respectively.

**Number of parameters**. Table 1 also exhibits the total number of parameters in training. The baseline models have the same training parameters as the inference stage, while the RefConv models have an extra amount of parameters *during training only*. However, as we only utilize the transformed weights for inference, *the inference parameter number of Re-parameterized RefConv model is identical to the baseline*, introducing completely no extra inference costs.

**Training-time FLOPs and memory costs**. To measure the extra training cost brought by the extra computations of RefConv during training, we present in Table 1 the total *training-time* FLOPs and memory costs of baselines and RefConv models, which are tested on four RTX 3090 GPUs with a total batch size of 256 and full precision (fp32). As exhibited, the additional FLOPs and memory that RefConv introduces is negligible compared to the baseline, complying with the discussion in Section 3.2 that the computational cost of Refocusing Transformation is minor since it is conducted on the kernels instead of the feature maps. It is worth noting that only the training-time RefConv needs minor extra computations to generate $W_t$, and the Re-parameterized RefConv model will be structurally identical to the baseline after converting the weights (as there will be no Refocusing Transformation during inference at all), introducing completely no additional memory or computational cost in inference.

### 4.2 COMPARISON WITH OTHER RE-PARAMETERIZATIONS

We compare RefConv with other data-independent re-parameterization methods on ImageNet, i.e. structural re-parameterization (SR, including ACB (Ding et al., 2019), RepVGGB (Ding et al., 2021c), and DBB (Ding et al., 2021b)) and weight re-parameterization (WR, including WN (Salimans & Kingma, 2016), CWN (Huang et al., 2017b), and OWN (Huang et al., 2018)). The baseline models are ResNet-18 and MobileNetv2. Note that all the inference models of these methods are the same as the baselines. As Table 2 shows, WR brings negligible improvement for that WR is intended to accelerate and stablize the training. While SR improves the performance more signifi-

Table 2: Comparison with other re-parameterization methods on ImageNet.

| Model | ResNet-18 | | | MobileNetv2 | | |
|---|---|---|---|---|---|---|
| Method | Top1-Accuracy | FLOPs (G) | Memory (G) | Top1-Accuracy | FLOPs (G) | Memory (G) |
| Baseline | 70.69% (+0.00%) | 472.08 | 15.52 | 71.68% (+0.00%) | 90.37 | 24.21 |
| ACB | 71.47% (+0.78%) | 761.25 | 18.76 | 71.99% (+0.31%) | 98.62 | 32.02 |
| RepVGGB | 71.21% (+0.52%) | 522.85 | 17.02 | 72.11% (+0.33%) | 94.39 | 28.38 |
| DBB | 71.25% (+0.56%) | 1097.40 | 26.18 | 72.25% (+0.48%) | 125.61 | 43.52 |
| WN | 70.81% (+0.12%) | 472.10 | 15.53 | 71.76% (+0.08%) | 90.37 | 24.21 |
| CWN | 70.85% (+0.16%) | 472.10 | 15.53 | 71.78% (+0.10%) | 90.37 | 24.21 |
| OWN | 70.83% (+0.14%) | 472.10 | 15.53 | 71.75% (+0.07%) | 90.37 | 24.21 |
| **RefConv** | **71.63% (+0.94%)** | 472.20 | 15.76 | **72.35% (+0.67%)** | 90.72 | 24.98 |

Table 3: Pascal VOC detection and Cityscapes segmentation.

| Task | Pascal VOC (mAP) | | Cityscapes (mIOU) | |
|---|---|---|---|---|
| Model | Baseline | RefConv | Baseline | RefConv |
| ResNet-18 | 68.76% | 69.48% (**+0.72%**) | 70.18% | 71.01% (**+0.83%**) |
| MobileNetv2 | 70.40% | 70.88% (**+0.48%**) | 72.31% | 72.93% (**+0.62%**) |
| MobileNetv3-L | 70.31% | 71.12% (**+0.81%**) | 72.04% | 72.86% (**+0.82%**) |

cant than WR, it introduces tremendous extra training costs since the extra branches are conducted on the feature maps. Furthermore, RefConv brings the highest improvements with little extra costs, demonstrating the superiority over other data-independent re-parameterization methods.

## 4.3 OBJECT DETECTION AND SEMANTIC SEGMENTATION

We further transfer the ImageNet-trained backbones to Pascal VOC detection task with SSD (Liu et al., 2016), following the configuration in Zhou et al. (2020), and Cityscapes segmentation with DeepLabv3+ (Chen et al., 2017), following the configuration in Ding et al. (2022b). Table 3 shows that RefConv can enhance the performance of various ConvNets by a clear margin, validating the transfer capability of RefConv.

## 4.4 ABLATION STUDY

**Refocusing Learning outperforms simply retraining the baseline model**. To show the superiority of Refocusing Learning over the most naive approach, which is simply training the model for more epochs, we train the already pre-trained baseline models for the second time on ImageNet with the same training configurations. Table 4 shows that retraining the models another time can barely improve the performance, which is expected as a specific kernel parameter during retraining still cannot attend to the other parameters at the other channels (for the case of DW conv in MobileNet) or spatial locations (for the case of regular conv in ResNet-18), thus failing to learn new representations.

**Refocusing Learning outperforms simply finetuning the baselines.** We also finetune the already pre-trained baseline models with a smaller learning rate of $10^{-4}$ for another 100 epochs on ImageNet as a common practice. Table 4 shows finetuning still brings negligible benefits, compared to RefConv. Once again, simply finetuning the converged models fail to learn any new representations.

**Pre-trained basis weights are important prior knowledge**. $W_b$ is the learned weights of the baseline models and fixed during Refocusing Learning. For validation, we randomly initialize the $W_b$ and freeze it in Refocusing Learning (Column $R\ W_b$ in Table 4). Doing so brings only minor improvements to MobileNetv2 and even results in significant degradation of ResNet-18, which is expected as the pre-trained basis weights can be regarded as prior knowledge brought into the Ref-Conv models, which provides a good basis for learning new representations. The phenomenon that ResNet-18 is degraded much worse can be explained that its Refocusing Transformation is a DW conv (as discussed in Section 3.2) that operates on the randomly initialized basis weights. Since the basis weights contain no priors at all, it is expected that the DW conv extracts no useful representations with only spatial aggregations. In contrast, the Refocusing Transformation of a DW RefConv in MobileNetv2 is a dense conv, which has a large parameter space to learn the representations all by itself, even if it begins with no priors.

Then we attempt to make the pre-trained $W_b$ trainable so that both $W_b$ and $W_r$ are updated in Refocusing Learning. Column $T\ W_b$ in Table 4 shows no improvements over the standard RefConv,

Table 4: Results of re-training and fine-tuning the baselines, and different configurations of the basis weights.

| Model | Baseline | RefConv | Retrain | Finetune | $R\ W_b$ | $T\ W_b$ | $R\&T\ W_b$ |
|---|---|---|---|---|---|---|---|
| ResNet-18 | 70.69% | **71.63%** | 70.85% | 70.75% | 53.04% | 71.54% | 71.24% |
| ↑ | +0.00% | **+0.94%** | +0.16% | +0.06% | -17.65% | +0.85% | +0.55% |
| MobileNetv2 | 71.68% | **72.35%** | 71.83% | 71.79% | 71.90% | 72.11% | 72.15% |
| ↑ | +0.00% | **+0.67%** | +0.15% | +0.11% | +0.22% | +0.43% | +0.47% |

Table 5: Results of different initialization of the trainable weights and the RefConv with/without the identity mapping.

| Model | Baseline | RefConv-RI | RefConv-ZI | RefConv w/o shortcut |
|---|---|---|---|---|
| MobileNetv1 | 72.18% | **72.96% (+0.78%)** | 72.89% (+0.71%) | 72.39% (+0.21%) |
| MobileNetv2 | 71.68% | **72.35% (+0.67%)** | 72.25% (+0.57%) | 71.89% (+0.21%) |
| MobileNetv3-S | 61.95% | **63.42% (+1.47%)** | 63.39%(+1.44%) | 62.95% (+1.00%) |
| MobileNetv3-L | 71.73% | **72.91% (+1.18%)** | 72.72% (+0.99%) | 72.27% (+0.54%) |

| Layer 2 | Layer 4 | Layer 6 | Layer 8 | Layer 10 | Layer 12 |

Figure 3: The connection degree matrix of the first 64 channels of $W_t$ and $W_b$ in different layers. The backbone model is MobileNetv1 trained on ImageNet. Darker colors represent larger values and closer connections.

suggesting that it is favorable to maintain the prior knowledge of the Refocusing Transformation. Last, we make $W_b$ both randomly initialized and trainable. Column $R\ \& \ T\ W_b$ in Table 4 shows performance lower than the standard RefConv. In summary, we conclude that the pre-trained basis weights $W_b$ are prior knowledge important to the learning process.

**Different initialization for refocusing weights**. The weights of CNNs are usually randomly initialized when training the models from scratch. However, for RefConv, $W_r$ can be initialized as zeros so that the initial value of $W_t$ will be identical to $W_b$ (by Eq. 1), making the initial RefConv Model equivalent to the pre-trained. Column *RefConv-ZI* in Table 5 shows the results of such zero initialization, demonstrating improvements over the baselines, which are slightly worse than the regular random initialization labeled as *RefConv-RI*.

**Validation of the identity mapping in RefConv**. We also discover that the identity mapping in RefConv is critical. Column *RefConv w/o shortcut* in Table 5 shows the results of RefConv without the shortcut, which are observably better than the baselines but worse than the standard RefConv.

**RefConv connects the independent kernels.** To validate that a DW RefConv makes each independent kernel channel of $W_t$ attend to the other channels of $W_b$, we calculate the degree of connection between the $i$-th channel $W_t$ and the $j$-th channel in $W_b$ for all the $(i, j)$ pairs, which forms a correlation matrix. Naturally, as such inter-channel connections are established through the filter $W_r^{(i,j)}$, which is a $k \times k$ matrix corresponding to the $j$-th input channel and $i$-th output channel of the $W_r$, we use the magnitude (i.e., the sum of the absolute values) of $W_r^{(i,j)}$ as the numerical metric for the degree of the connection, as a common practice (Han et al., 2015b; Ding et al., 2018; Li et al., 2016; Guo et al., 2016; Ding et al., 2019). Briefly, a larger magnitude value indicates a stronger connection. We use the first 64 channels of $W_t$ and $W_b$ and calculated the connection degree between each pair of channels to obtain the $64 \times 64$ connection degree matrix. As visualized in Fig. 3, the $i$-th channel in $W_t$ attends to not only the corresponding $i$-th channel in $W_b$ but also multiple other channels in $W_b$ with different magnitude, suggesting that a DW RefConv can attend to all the channels to learn diverse combinations of existing representations.

## 4.5 RefConv Reduces Channel Redundancy

To explore the difference between the basis weights $W_b$ and the transformed weights $W_t$, we compare the channel redundancy of $W_b$ and $W_t$. As a common practice, we utilize the Kullback-Leibler

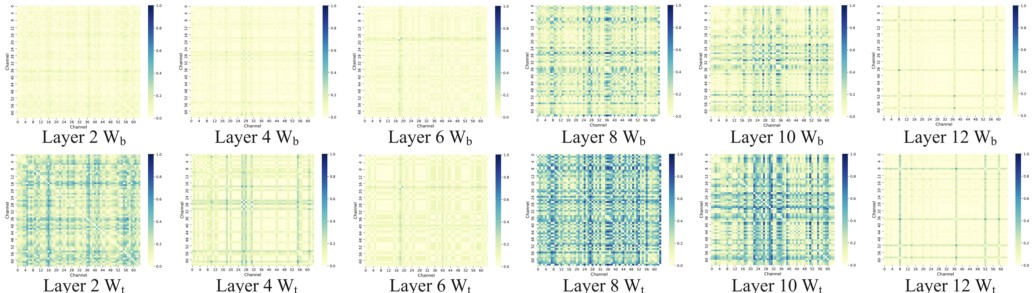

Figure 4: The similarity matrix of the first 64 channels of $W_b$ and $W_t$ in different layers. The backbone model is MobileNetv1 trained on ImageNet. To improve the readability, the original value of KL divergence is added with 1 and then taken 10-base logarithm. A point with a darker color represents a larger value, hence a lower similarity.

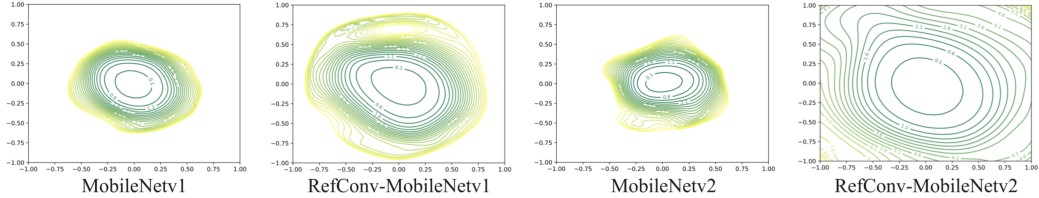

Figure 5: Visualization of the loss landscapes.

(KL) divergence to measure the similarity between different pairs of channels (Zhou et al., 2019; Wang & Stella, 2021), so that a larger KL divergence indicates lower similarity, hence a lower degree of channel redundancy. Specifically, we sample a DW RefConv layer from the trained MobileNetv1 and apply softmax to every $K \times K$ kernel channel, then sample the first 64 channels to calculate the KL divergence between every pair of channels. In this way, we obtain a $64 \times 64$ similarity matrix for the sampled layer. Fig. 4 shows the similarity matrices of multiple layers. As can be observed, there exists high redundancy among channels in $W_b$ as the KL divergence is low between most of the channels. In contrast, the KL divergence between channels of $W_t$ is significantly higher, which means the kernel channels become significantly different from the others. Based on such observations, we conclude that RefConv can reduce redundancy consistently and effectively. We explain such phenomena that RefConv can explicitly make every channel able to attend to the other channels of the pretrained kernel, which refocus on the learned representations encoded in the pretrained kernel channels to learn diverse novel representations. Consequently, the channel redundancy is reduced and the representation diversity is enhanced, which results in a higher representational capacity.

### 4.6 REFCONV SMOOTHS LOSS LANDSCAPE

To explore how Refocusing Learning influences the training dynamics, we visualize the loss landscapes of the baseline and the RefConv counterpart with the filter-wise normalization visualization (Li et al., 2018). We use MobileNetv1 and MobileNetv2 trained on CIFAR-10 as the backbone models. Fig. 5 shows that the loss landscapes of RefConv have wider and sparser contours compared to that of the baselines, which indicates that the loss curvature of RefConv is much flatter (Li et al., 2018), suggesting a better generalization ability. This phenomenon demonstrates that Refocusing Learning possesses better training properties which partly explains the performance improvements.

## 5 CONCLUSION

This paper proposes Re-parameterized Refocusing Convolution (RefConv), which is the first re-parameterization method that augments the priors of existing model structures by establishing extra connections among kernel parameters. As a plug-and-play module to replace the regular convolutional layers, RefConv can significantly improve the performance of various CNNs on multiple tasks without altering the original model structures or introducing extra costs in inference. Moreover, we explain the effectiveness of RefConv by showing its capability of reducing channel redundancy and smoothing the loss landscape, which may inspire further theoretical research on training dynamics. In our future work, we will explore more effective designs of Refocusing Transformations, e.g., by introducing non-linearity and more advanced operations.

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

APPENDIX A: REFCONV STRENGTHENS THE KERNEL SKELETONS

To explore the other differences between the basis kernels and transformed kernels, we visualize the $W_b$, $W_t$ and the increments over the basis weights $\Delta W = W_b - W_t$ of the last convolution layer of RefConv-MobileNetv2 trained on ImageNet, as exhibits in the left column of Fig. 6. We find that most of the $\Delta W$ exhibits stronger skeleton patterns (Ding et al., 2019), indicating that the major difference lies in the center rows and columns of the kernels. Consequently, it can be obviously observed that $W_t$ exhibits stronger skeleton patterns than $W_b$, especially in the central point. Further more, we calculate and visualize the average kernel magnitude matrices (Ding et al., 2019; 2018; Guo et al., 2016; Han et al., 2015b;a)of these three weights, as exhibits in the right column of Fig. 6. Once again, the magnitude of $\Delta W$ shows strong skeleton patterns and small impact factors in corners, suggesting the skeleton patterns of $W_t$ are strengthened and the corners are weakened, compared to $W_b$. Moreover, it is noteworthy that for $W_t$, the central point has a value of 1.000, which means that location has a dominant importance consistently in every $3 \times 3$ layers. According to Ding et al. (2019), enhancing the skeletons results in performance improvement, which explains the effectiveness of RefConv from another perspective.

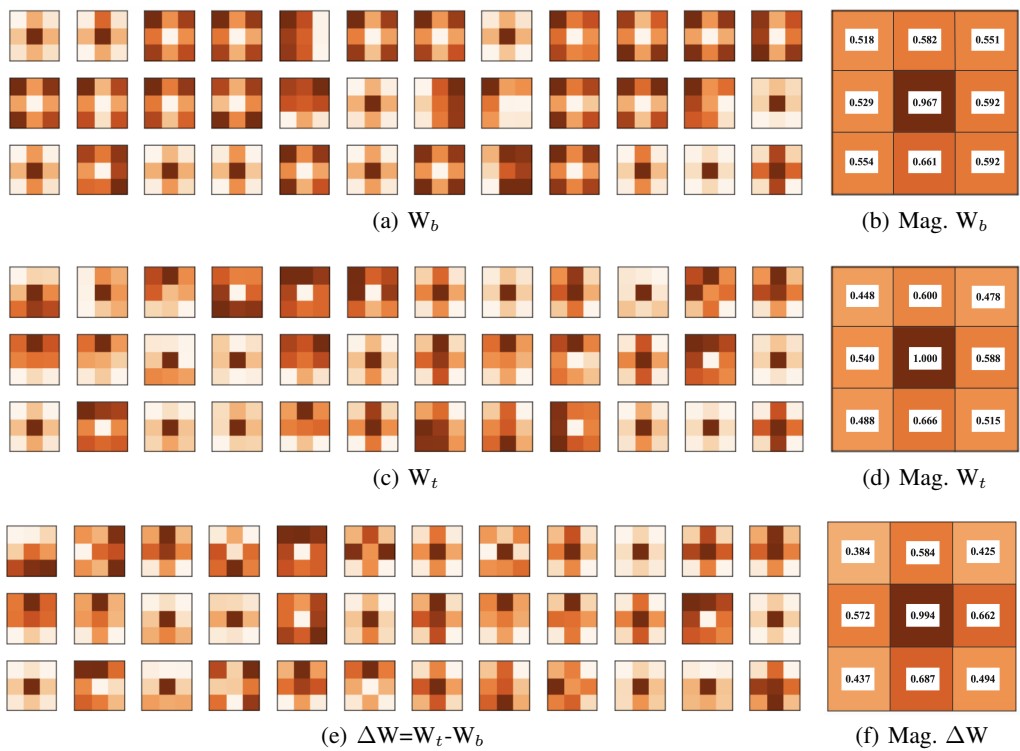

(a) $W_b$                (b) Mag. $W_b$

(c) $W_t$                (d) Mag. $W_t$

(e) $\Delta W = W_t - W_b$                (f) Mag. $\Delta W$

Figure 6: Visualizations of the magnitude (i.e., the absolute value of weights) of individual kernel channels and the average magnitude matrices (which are averaged across channels). The weights are sampled from the last RefConv layer of the RefConv-MobileNetv2 trained on ImageNet. For the visualization, we normalize each matrix by the maximal value of its entries to facilitate the comparison and improve the readability. A darker color indicates a larger magnitude.

APPENDIX B: PERFORMANCE EVALUATION ON OTHER DATASETS

**Evaluation of RefConv on Other Datasets.** We also test the effectiveness of RefConv on CIFAR-10, CIFAR-100, and Tiny-ImageNet-200. We resize the images to $224 \times 224$, and the training strategy is in accordance with the experiments on ImageNet. A set of representative CNN architectures are tested, including Cifar-quick (Krizhevsky et al., 2012), SqueezeNet (Iandola et al., 2016), VG-GNet (Simonyan & Zisserman, 2014), ResNet (He et al., 2016), ShuffleNetv1,v2 (Zhang et al., 2018;

Table 6: Results of RefConv and the normally trained baselines on CIFAR-10, CIFAR-100 and Tiny-ImageNet-200.

| Dataset | CIFAR-10 | | CIFAR-100 | | Tiny-ImageNet-200 | |
|---|---|---|---|---|---|---|
| Model | Baseline | RefConv | Baseline | RefConv | Baseline | RefConv |
| Cifar-quick | 83.02% | 89.13% (**+6.11%**) | 55.21% | 61.06% (**+5.85%**) | 56.90% | 60.24% (**+3.34%**) |
| SqueezeNet | 85.83% | 86.64% (**+0.81%**) | 61.14% | 61.94% (**+0.80%**) | 51.60% | 53.04% (**+1.44%**) |
| VGGNet-16 | 92.50% | 93.13% (**+0.63%**) | 73.75% | 74.88% (**+1.13%**) | 61.88% | 63.52% (**+1.64%**) |
| ResNet-18 | 93.10% | 94.15% (**+1.05%**) | 73.65% | 74.51% (**+0.86%**) | 61.12% | 62.44% (**+1.32%**) |
| ResNet-34 | 93.95% | 94.97% (**+1.02%**) | 74.82% | 95.74% (**+0.92%**) | 65.21% | 66.45% (**+1.24%**) |
| ShuffleNetv1 | 91.35% | 92.69% (**+1.34%**) | 68.51% | 69.78% (**+1.27%**) | 56.86% | 58.62% (**+1.76%**) |
| ShuffleNetv2 | 92.31% | 93.56% (**+1.25%**) | 70.08% | 71.52% (**+1.44%**) | 60.38% | 61.90% (**+1.52%**) |
| ResNeXt-18 | 93.00% | 93.68% (**+0.68%**) | 72.02% | 72.84% (**+0.82%**) | 62.02% | 62.84% (**+0.82%**) |
| RegNetX_200MF | 91.11% | 91.93% (**+0.82%**) | 68.08% | 69.73% (**+1.65%**) | 59.84% | 61.23% (**+1.39%**) |
| ResNeSt-50 | 93.52% | 95.22% (**+1.70%**) | 69.18% | 70.82% (**+1.64%**) | 65.18% | 65.82% (**+0.64%**) |
| FasterNet-T0 | 92.94% | 94.08% (**+1.14%**) | 68.02% | 69.24% (**+1.22%**) | 58.32% | 60.14% (**+1.82%**) |
| MobileNetv1 | 92.45% | 93.01% (**+0.56%**) | 72.43% | 73.41% (**+0.98%**) | 62.52% | 63.89% (**+1.37%**) |
| MobileNetv2 | 92.58% | 93.71% (**+1.13%**) | 73.04% | 74.35% (**+1.31%**) | 61.94% | 63.96% (**+2.02%**) |
| MobileNetv3-S | 90.91% | 92.23% (**+1.32%**) | 68.20% | 70.76% (**+2.56%**) | 60.66% | 62.71% (**+2.05%**) |
| MobileNetv3-L | 92.95% | 93.89% (**+0.94%**) | 73.89% | 75.27% (**+1.38%**) | 62.21% | 64.28% (**+2.07%**) |
| MobileNeXt | 93.01% | 94.45% (**+1.44%**) | 67.42% | 69.07% (**+1.65%**) | 58.12% | 60.44% (**+2.32%**) |
| HBONet | 92.32% | 93.59% (**+1.27%**) | 72.39% | 73.91% (**+1.52%**) | 64.20% | 66.44% (**+2.24%**) |
| EfficientNetv1-B0 | 94.21% | 95.38% (**+1.17%**) | 75.68% | 76.90% (**+1.22%**) | 66.62% | 68.57% (**+1.95%**) |
| EfficientNetv2-S | 93.85% | 95.13% (**+1.28%**) | 75.42% | 76.76% (**+1.34%**) | 67.18% | 69.32% (**+2.15%**) |

Table 7: Comparison with other re-parameterization methods on ImageNet, CIFAR-10, CIFAR-100 and Tiny-Image-200.

| Method | | CIFAR-10 | CIFAR-100 | Tiny-ImageNet-200 |
|---|---|---|---|---|
| Baseline Model | ResNet-18 | 93.10% (+0.00%) | 73.65% (+0.00%) | 61.12% (+0.00%) |
| Structural Rep | ACB | 94.03% (+0.93%) | 74.45% (+0.80%) | 61.98% (+0.86%) |
| | RepVGGB | 93.59% (+0.49%) | 73.98% (+0.33%) | 61.78% (+0.66%) |
| | DBB | 93.81% (+0.71%) | 73.96% (+0.31%) | 62.12% (+1.00%) |
| Weight Rep | WN | 93.49% (+0.39% ) | 74.21% (+0.56%) | 61.58% (+0.46%) |
| | CWN | 93.79% (+0.69%) | 74.19% (+0.54%) | 61.74% (+0.62%) |
| | OWN | 93.85% (+0.75%) | 74.27% (+0.62%) | 61.64% (+0.52%) |
| **Rep Refocusing** | **RefConv** | **94.15% (+1.05%)** | **74.51% (+0.86%)** | **62.44% (+1.32%)** |
| Baseline Model | MobileNetv2 | 92.58% (+0.00%) | 73.04% (+0.00%) | 61.94% (+0.00%) |
| Structural Rep | ACB | 93.39% (+0.81%) | 73.68% (+0.64%) | 62.82% (+1.08%) |
| | RepVGGB | 93.21% (+0.63%) | 73.82% (+0.78%) | 62.66% (+0.72%) |
| | DBB | 93.46% (+0.88%) | 74.02% (+0.98%) | 63.44% (+1.50%) |
| Weight Rep | WN | 92.83% (+0.25%) | 73.33% (+0.29%) | 62.04% (+0.10%) |
| | CWN | 92.81% (+0.23%) | 73.31% (+0.27%) | 62.07% (+0.13%) |
| | OWN | 92.85% (+0.27%) | 73.37% (+0.23%) | 62.12% (+0.18%) |
| **Rep Refocusing** | **RefConv** | **93.71% (+1.13%)** | **74.35% (+1.31%)** | **63.96% (+2.02%)** |

Ma et al., 2018), ResNeXt (Xie et al., 2017), RegNet (Radosavovic et al., 2020), ResNeSt (Zhang et al., 2020a), FasterNet (Chen et al., 2023), MobileNetv1,v2,v3 (Howard et al., 2017; Sandler et al., 2018; Howard et al., 2019), MobileNeXt (Zhou et al., 2020), HBONet (Li et al., 2019a), and EfficientNetv1,v2 (Tan & Le, 2019; 2021). The results are shown in Table 6. As can be observed, the performance of all models is consistently improved by a clear margin. For example, RefConv significantly enhances the performance of Cifar-quick (with dense conv) by 6.11%, 5.85% and 3.34% on CIFAR-10, CIFAR-100 and Tiny-ImageNet-200 respectively. Besides, RefConv improves the performance of ShuffleNetv1 (with group-wise conv) by 1.34%, 1.27% and 1.76% on CIFAR-10, CIFAR-100 and Tiny-ImageNet-200 respectively. As for the DW Conv, RefConv improves the top-1 accuracy of MobileNetv3-S by 1.32% 2.56% and 2.05% on CIFAR-10, CIFAR-100 and Tiny-ImageNet-200 respectively. In summary, the results validate that RefConv can enhance the repre-

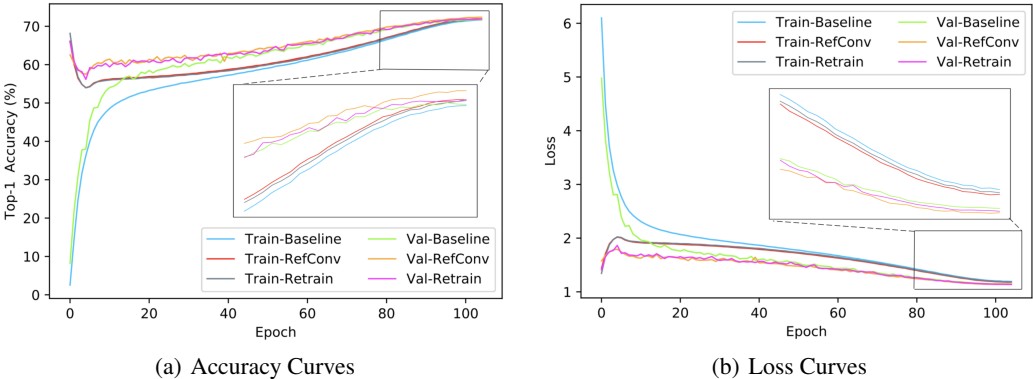

(a) Accuracy Curves  (b) Loss Curves

Figure 7: Training and validation of Top-1 accuracy and loss curves on ImageNet classification of MobileNetv2, MobileNetv2-retrain, and RefConv-MobileNetv2.

sentational capacity of various models with different types of convolution layers, including dense conv, group-wise conv and DW conv.

**Comparison with Other Re-parameterizations on Other Datasets.** We further compare Re-fConv with other data-independent re-parameterization methods on CIFAR-10, CIFAR-100, and Tiny-ImageNet-200. We also resize the images to $224 \times 224$, and follow the training strategy on ImageNet. The base models are ResNet-18 and MobileNetv2. The tested structural re-parameterization methods are ACB (Ding et al., 2019) , RepVGG Block (Ding et al., 2021c) and DBB (Ding et al., 2021b). The tested data-independent weight re-parameterizations methods are Weight Normalization (WN) (Salimans & Kingma, 2016), Centered Weight Normalization (CWN) (Huang et al., 2017b) and OWN (Huang et al., 2018). As Table 7 shows that Refocusing Learning brings the highest improvements, demonstrating the superiority of RefConv over the other re-parameterization methods.

## APPENDIX C: TRAINING DYNAMICS OF REFCONV

Fig. 7 shows the curves of loss and accuracy on the training and validation datasets of the baseline, retrained (as defined in the paper), and the RefConv counterpart of MobileNetv2, which are trained on ImageNet in 100 epochs with a 5-epoch warmup. Comparing RefConv with the baseline, we observe that the training of the RefConv model converges faster than the baseline. Moreover, the model with RefConv consistently has a higher training/validation accuracy and a lower training/validation loss than the baseline during the optimization process. As reported in the paper, MobileNetv2 with RefConv finally converges to a better state than the baseline as revealed by the higher validation accuracy. In addition, we find that the baseline starts from low accuracy and ascends rapidly in the first about 15 epochs, and then ascends relatively smoothly. In contrast, the RefConv model starts with a relatively high accuracy (this is because RefConv conducts on the basis of the pre-trained model), and its accuracy declines slightly in the beginning and then smoothly ascends. For the loss curves, we find that the loss curve of the baseline starts from a high value and declines rapidly in the first about 15 epochs, and then declines relatively smoothly. In contrast, the loss of the RefConv model starts from a relatively low value and ascends slightly in the beginning, then consistently declines smoothly. Such observations suggest that the RefConv model has totally different training dynamics from the baseline.

We use the retrained model (labeled as *Retrain*) for another set of comparisons since it also begins with weights inherited from the baseline model. Comparing the RefConv and retrained models with the retrain, we observe that their accuracy/loss curves have similar appearances. However, we find that the accuracy of the RefConv model starts from a lower value, and the loss starts higher. This is expected since RefConv introduces an extra set of randomly initialized learnable parameters, learning to establish new connections among the basis parameters and generating new kernels. In contrast, the optimization of the retrained model starts precisely from the ending point of the optimization of the baseline model. Therefore, the retrained model has a higher starting accuracy and a

Table 8: Results of baselines, RefConv, RefConv with half training data on CIFAR-10 and Tiny-ImageNet-200.

| Dataset | CIFAR-10 | | | Tiny-ImageNet-200 | | |
|---|---|---|---|---|---|---|
| Model | Baseline | RefConv | Half-RefConv | Baseline | RefConv | Half-RefConv |
| ResNet-18 | 93.10% | 94.15% | 94.11% (-0.04%) | 61.12% | 62.44% | 62.39% (-0.05%) |
| ResNet-34 | 93.95% | 94.97% | 94.95% (-0.02%) | 65.21% | 66.45% | 66.41% (-0.04%) |
| ShuffleNetv1 | 91.35% | 92.69% | 92.66% (-0.03%) | 56.86% | 58.62% | 58.56% (-0.06%) |
| ShuffleNetv2 | 92.31% | 93.56% | 93.51% (-0.04%) | 60.38% | 61.90% | 61.86% (-0.04%) |
| MobileNetv1 | 92.45% | 93.01% | 93.03% (+0.02%) | 62.52% | 63.89% | 63.85% (-0.04%) |
| MobileNetv2 | 92.58% | 93.71% | 93.70% (-0.01%) | 61.94% | 63.96% | 63.94% (-0.02%) |
| MobileNetv3-S | 90.91% | 92.23% | 92.12% (-0.11%) | 60.66% | 62.71% | 62.67% (-0.04%) |
| MobileNetv3-L | 92.95% | 93.89% | 93.91% (-0.02%) | 62.21% | 64.28% | 64.22% (-0.06%) |

lower loss but the resultant performance is inferior to the RefConv model since no novel connections are established.

## APPENDIX D: REFOCUSING LEARNING WITH PART OF THE TRAINING DATA

We wonder whether Refocusing Learning still works when it only has access to part of the training data instead of the whole training set. Thus we train RefConv with half training set of CIFAR-10 and Tiny-ImageNet-200, and evaluation on the whole test set. We follow the optimization strategy on ImageNet as stated above except for that we maintain the number of iterations through halving the batch-size from 256 to 128. Since Refocusing Learning only accesses to half of the training data, the total training costs are also halved. As column *Half-RefConv* in Table 8 shows, RefConv can still achieve satisfying performance under this setting.

## APPENDIX E: CODE IN PYTORCH

RefConv is simple to implement on the mainstream CNN frameworks like PyTorch. We provide the PyTorch-like code of RefConv in Algorithm 1. When implementing on CNN models, we only need to replace the non-pointwise Conv layers in the model with RefConv. The implementation code of RefConv on CNNs in PyTorch is available in the Supplemental Materials.

**Algorithm 1** RefConv, PyTorch-like code.

```python
import torch.nn as nn
from torch.nn import functional as F

class RefConv(nn.Module):

    """
    Implementation of RefConv.
    --in_channels: number of input channels in the basis kernel
    --out_channels: number of output channels in the basis kernel
    --kernel_size: size of the basis kernel
    --stride: stride of the original convolution
    --padding: padding added to all four sides of the basis kernel
    --groups: groups of the original convolution
    --map_k: size of the learnable kernel
    """

    def __init__(self, in_channels, out_channels, kernel_size, stride,
    padding=None, groups=1, map_k=3):
        super(RefConv, self).__init__()

        assert map_k <= kernel_size
        self.origin_kernel_shape = (out_channels, in_channels // groups,
        kernel_size, kernel_size)
        self.register_buffer('weight', torch.zeros(*self.origin_kernel_shape))
        G = in_channels * out_channels // (groups ** 2)
        self.num_2d_kernels = out_channels * in_channels // groups
        self.kernel_size = kernel_size
        self.convmap = nn.Conv2d(in_channels=self.num_2d_kernels,
                            out_channels=self.num_2d_kernels,
                            kernel_size=map_k, stride=1, padding=map_k // 2,
                            groups=G, bias=False)
        #nn.init.zeros_(self.convmap.weight)
        #zero initialization the trainable weights
        self.bias = None
        #nn.Parameter(torch.zeros(out_channels), requires_grad=True)
        self.stride = stride
        self.groups = groups
        if padding is None:
            padding = kernel_size // 2
        self.padding = padding

    def forward(self, inputs):

        origin_weight = self.weight.view(1, self.num_2d_kernels, self.kernel_size,
            self.kernel_size)
        kernel = self.weight + self.convmap(origin_weight).view(*self.
            origin_kernel_shape)
        return F.conv2d(inputs, kernel, stride=self.stride, padding=self.padding,
            dilation=1, groups=self.groups, bias=self.bias)
```

