# OpenReview forum: "RefConv: Re-parameterized Refocusing Convolution for Powerful ConvNets"
_ICLR.cc/2024/Conference — Submitted to ICLR 2024_

### Official Review · Reviewer_qHhh · 2023-10-31

**Soundness:** 4 excellent
**Presentation:** 3 good
**Contribution:** 3 good
**Rating:** 8
**Confidence:** 4

**Summary:**

This paper proposed a novel re-parameterization method named Re-parameterized Refocusing, which can establish connections across the channels of the learned conv kernel.
Experiments show that the proposed method can improve the performance of many convnets in various tasks, such as image classification and segmentation, without introducing any computation cost in inference phase.

**Strengths:**

1. The proposed method is novel and effective.
2. The experiments are solid.

**Weaknesses:**

None

**Questions:**

None

---

### Official Review · Reviewer_Z4v5 · 2023-10-31

**Soundness:** 2 fair
**Presentation:** 2 fair
**Contribution:** 2 fair
**Rating:** 5
**Confidence:** 5

**Summary:**

The paper introduces a novel approach to neural network training, specifically by bifurcating the weight training process into two distinct stages. However, I believe the validation of the method's effectiveness is not adequately comprehensive. Given the current state of the paper, my recommendation would be to not accept it in its present form.

**Strengths:**

The concept presented in the paper captures the interest.


Figure 1 is exceptionally clear and effectively conveys the central concept of the method proposed in the paper.


Because the parameters are seamlessly integrated, the proposed method does not incur additional costs during the inference phase.


The paper employs techniques such as visualization to offer numerous valuable insights.

**Weaknesses:**

The proposed method incurs a higher training cost compared to the original approach. My concern does not lie with the cost itself; rather, I am questioning the accuracy and reliability of the validation process employed.


The authors believe that their method can indirectly connect information from different channels of the input, which is clearly a mistake. Let x = [x1, x2, ..., xc]; y = [y1, y2, ..., yc]. It is obvious that y1 does not contain the content from x2 to xc. If there is any, please prove it using the notation I provided.



I am quite familiar with ImageNet, and I have concerns about the data presented in Table 1. I would like the authors to refer to the data from timm. The authors might argue that their values are lower than the standard libraries in timm because they only trained for 100 epochs, but I consider this a drawback. If the standard training procedure from timm was used, perhaps the authors' method would not show any gain. It is conceivable that the authors' method is essentially equivalent to providing more extensive training to an originally under-trained model, albeit with a longer training time. If a model is adequately and standardly trained, the authors' method should be unnecessary.


In the first experiment of Section 4.4, the authors should train all models to full convergence (for instance, more than 500 epochs) before making comparisons. Stepping back, when the authors retrain this model, do they use twice the training epochs?



The second experiment in Section 4.4 is incorrect. The authors should not simply use a small learning rate to fine-tune; instead, they should follow timm’s practice of training from scratch for 500 epochs until convergence.

**Questions:**

See #Weakness.

---

> ### Author Response · Authors · 2023-11-22
>
> We sincerely thank the reviewer for appreciating the conceptual novelty and insights of our paper. We have revised the paper according to the constructive feedbacks.
>
> Weakness 1 (connecting the channels)
>
> Given x = [x1, x2, ..., xc]; K=[K1, K2, ..., Kc]; y = [y1, y2, ..., yc], it is true that K1 can not **directly** extract the features of x2 to xc. However, in the pre-training stage, each Ki is trained to extract the feature patterns of xi, that is to say, each Ki learns certain representations corresponding to xi. Let function R represent the pre-training process that produces the channel weights given the features of a channel, we denote that by Ki:=R(xi) because yi=conv2d(Ki, xi) in the pretraining phase (conv2d is the convolution operation). RefConv makes each kernel channel learn a combination of every channel through the trainable Refocusing Transformation T, namely, New_Ki = T(K1, K2, ..., Kc) = T(R(x1), R(x2), ..., R(xc)), thus RefConv essentially makes the transformed kernel learn the combination of representations of each input feature channel xi. With such a kernel New_K, we have new_yi = conv2d(New_Ki, xi) = conv2d(T(R(x1), R(x2), ..., R(xc)), xi), which **indirectly** establishes the connections between different channels.
>
> We explained this mechanism in the original paper (the middle part of Page 2). [With a properly designed Refocusing Transformation, we can relate the parameters of a specific kernel channel to the parameters of the other kernel channels, i.e., make them refocus on the other parts of the model (rather than the input features only) to learn new representations. As the latter are trained with the other feature channels, they encode the representations condensed from the other feature channels, so that we can indirectly establish connections among the feature channels, which cannot be realized directly (by the definition of DW conv).]

---

> > ### Author Response · Authors · 2023-11-22
> >
> > Weakness 2 (regarding the training configurations and fairness of comparisons)
> > >If a model is adequately and standardly trained, the authors' method should be unnecessary\
> > >They should follow timm’s practice of training from scratch for 500 epochs until convergence.
> >
> > First, we would like to note that training for about 100 epochs is a common practice, which has been used by many classic and latest works [1-5].\
> > [1] Deep Residual Learning for Image Recognition. (CVPR 2016)\
> > [2] MobileNetV2: Inverted Residuals and Linear Bottlenecks. (CVPR 2018)\
> > [3] EfficientNet: Rethinking Model Scaling for Convolutional Neural Networks. (ICML 2019)\
> > [4] DyRep: Bootstrapping Training with Dynamic Re-parameterization (CVPR 2022)\
> > [5] Online Convolutional Re-parameterization (CVPR 2022)
> >
> > Second, as suggested, we report the results supporting the effectiveness of RefConv on the TIMM-pretrained ResNet-34 and MobileNetv3. We use these two models because they are the largest models we can afford to train for 1000 epochs before the author response deadline with our limited resources. We directly use the codebase provided in TIMM and its pre-trained weights. In the procedure of refocusing learning, we follow the original training script of the trained models to train for another 500 epochs. For the baseline models, we also train for another 500 epochs. \
> > On ResNet34, RefConv enhances the accuracy from 75.12% to 75.86%, while the accuracy of fine-tuning the pre-trained ResNet-34 is 75.16%.\
> > On MobileNetv3, RefConv enhances the accuracy from 74.28% to 74.91%, while the accuracy of fine-tuning is 74.34%.\
> > Such results demonstrate that even for the well-trained models, RefConv can also bring significant improvement.
> >
> > We have added such results into the appendix of the revised paper.
> >
> > >Stepping back, when the authors retrain this model, do they use twice the training epochs?
> >
> > **The answer is yes**, but first, we would like to clarify that even without extra training epochs, RefConv can still bring accuracy improvement. As shown in the last experiments of paragraph **Pre-trained basis weights are important prior knowledge** in Section 4.4, we use a randomly initialized baseline (namely, no pre-trained models are provided), and train both of the W_b and W_r simultaneously for 100 epochs, namely, the total training epochs are 100, and we use no extensive training epochs here. Under this setting, RefConv can also enhance the performance of the baseline models, as shown in column R&T W_b of Table 4.\
> > This experiment proves that the fact that RefConv improves the model's performance cannot be simply attributed to the more training epochs (otherwise RefConv would be ineffective in this joint learning). However, once given a trained model, RefConv is the only usable and effective re-parameterization method for further performance improvement.
> >
> > Then, we would like to emphasize that even given the same total training epochs, the accuracy of the baseline models fails to be improved, while RefConv is the only method bringing accuracy enhancement. We conduct experiments that train the baselines for doubled epochs (retrain and fine-tune) in total as shown in Section 4.4. Both retraining and fine-tuning the trained baselines, which both use double epochs (i.e., the same number of epochs as RefConv), fail to bring any significant performance improvement as shown in Table 4, thus validating that the model converged and conducting more epochs of training does not improve the performance. Such results also suggest that simply extending the training schedule may not be the key to the high performance of TIMM models as they also use modern data augmentations and training tricks.\
> > In summary, we would like to note that RefConv, retrain, and fine-tune have the same total training epochs (200 epochs in total). However, RefConv brings enhancement while retrain and fine-tune fail to, which validates that the effectiveness of RefConv is not equivalent to providing more extensive training (otherwise retrain and fine-tune will be helpful), but comes from its unique mechanism of connecting the kernel channels.
> >
> > Thank you again for appreciating our work, and we would be grateful if you could raise the score.
> > Please let us know if there is anything we can do to convince you to further raise the score.

---

### Official Review · Reviewer_3dDh · 2023-11-09

**Soundness:** 3 good
**Presentation:** 4 excellent
**Contribution:** 2 fair
**Rating:** 5
**Confidence:** 5

**Summary:**

In this paper, the authors propose a technique called Re-parameterized Refocusing Convolution, which is based on the idea of structural re-parameterization, i.e., incorporating more learnable parameters into the model during training and training them for better performance. These parameters are merged into the original model's parameters during inference to achieve the goal of not introducing additional inference costs.

**Strengths:**

The approach in this paper can be viewed as "convolution B of convolution A", where A is the convolution parameter trained by the pre-training process and kept frozen once trained.B is the convolution parameter that continues to be trained.

The method in this paper has a slight advantage over several other structural reparameterization and weight reparameterization methods in terms of results.

I think the conclusion of the final analysis, "Re-parameterized refocusing reduces redundancy between channels", demonstrates well the changes that the methods in this paper can make to a pre-trained convolutional model.

The experiments included a variety of convolutional models.

**Weaknesses:**

Observe that the ImageNet experimental results have about 1% performance improvement on many models, but also a lot more Params.

The network architectures that come into play are generally early CNN models such as ResNet, DenseNet, MobileNet family, etc. For modern convolutional architectures such as SlaK, RepLKNet, HorNet, etc., the effect is currently unknown.

**Questions:**

1	The refocusing technique seems to be one that can be iterated. Can the refocusing technique in this paper continue to be iterative? I.e., after doing one refocusing exercise, then the next one. Will the results continue to improve?

2	For the base weight W_b, one of the points claimed by the refocusing technique is the possibility of establishing links between its individual channels. Why is this necessary? Each channel of the base weight kernel has its own role, so to link them?

3	During refocus training, the result after convolution of the base weight W_b with its previous features can be seen as a new "feature". The transform weight W_t can be seen as trainable to process this new "feature". This process is equivalent to fine-tuning the convolution after "injecting" new parameters. I would like to ask if any experiments with other models (e.g. new convolutional networks) have found that this degrades performance?

---

> ### Author Response · Authors · 2023-11-15
>
> We sincerely thank the reviewer for the constructive comments.
>
> Weakness 1 (more params in training)
>
> Although RefConv introduces extra params, these params are conducted on the basis weights instead of the feature maps, thus the extra Flops and memory cost introduced in training are negligible as stated in Sec.3.2 and quantitively shown in Table 1.
> Moreover, these extra params can be equivalently transformed after training, thus no extra params are introduced in inference.
>
> Weakness 2 (modern architectures)
>
> We also test the effectiveness of RefConv on two modern architectures, ConvNeXt [1] and FasterNet [2], the improvements are 0.96% and 1.15%, as shown in Table 1.\
> [1] A convnet for the 2020s. (CVPR 2022)\
> [2] Run, don’t walk: Chasing higher flops for faster neural networks. (CVPR 2023)
>
> Question 1
>
> As requested, we conduct refocusing technique for the second time on the trained MobileNetv2 with RefConv, the accuracy is slightly  enhanced from 72.35% to 72.43%. Then the accuracy remains close to 72.43% when more iterations of refocusing technique are conducted.\
> Thus the performance improvement that RefConv brings may have an upper limit, which may be because conducting the refocusing technique once has been capable enough to effectively reduce the channel redundancy and learn new representations.
>
> Question 2
>
> DW conv was originally proposed to reduce the model size and FLOPs but at the cost of representational capacity. Consider a two-channel input feature map whose shape is (2, H, W), assume the conv layer is a two-channel DW conv whose kernel shape is (2, 1, K, K). The first channel of output only relates to the first channel of the input, and the second channel of output only relates to the second channel of input. But if the kernel is a regular (dense) conv, the kernel shape will be (2, 2, K, K), so that each of the output channel relates to all the input channels. From this perspective, we can see that a regular conv's representational is higher than a DW conv, i.e., it comes at a cost to let "each channel of the base weight kernel have its own role".
>
> As evidence, in practice, if we only consider the performance, regular conv is always better than DW conv (namely, the accuracy will become higher when simply replacing DW conv with regular conv). Therefore, it's expected that implicitly connecting the channels of the DW conv can narrow the representational gap between the DW conv and the regular conv.
>
> We explain how RefConv works. Taking the aforementioned two-channel DW conv for example, we denote its input, output, and kernel by X, Y, and W, respectively, so that using pytorch-style pseudo code, this layer can be represented by\
> Y = conv2d(X, W)
>
> RefConv can link the channels by re-parameterizing the kernel and such a re-parameterization is realized with a convolution on W. So that\
> Y = conv2d(X, conv2d(W_b, W_r))\
> Note that conv2d(W_b, W_r) can be seen as a re-parameterization of the original W. Since conv2d(W_b, W_r) is a dense conv, each channel of the output directly relates to each channel of W_b, so that each output channel of Y **indirectly** relates to each channel of W_b. Note that we omit the "identity mapping" (Eq.1 in the paper) here for simplicity.
>
> Except for the improvements in performance, in this paper, we have also provided evidence that linking the channels brings benefits. In a DW conv, since there are no connections between the channels of DW conv, it is easy to result in redundancy. The evidence is shown in Section 4.5. In contrast, RefConv reduces the redundancy, as shown in Figure 4. This is because by linking the channels with W_r, the re-parameterized kernel W_t is encouraged to learn new representations through the learnable combination of the channels of W_b, which were originally independent.
>
> Question 3
>
> So far we have not encountered the performance degradation.\
> As discussed above, in the procedure of refocus training, the basis weights W_b dose not conduct convolution on the feature maps, it actually act as the input"features" of the refocusing transformation with W_r. Then the output generated by the refocusing transformation is the transformed weights W_t, which replaces the W_b and conducts the convolution on the feature maps. To conclude, in refocus training, W_b does not serve as a kernel to convolve on anything, W_r conducts refocusing transformation on W_b and generates W_t, and W_t is the kernel of convolution on the input feature maps and generates the output feature maps.\
> We reckon that RefConv is more than just simply fine-tuning the original conv layer. It uses learnable transformations to establish the connections between convolution kernels, learning new representations through combining the learned representations by the basis kernels, thus strengthening the model representational capacity.\
> Moreover, due to the use of "identity mapping" (Eq.1 in the paper), the representational capacity of W_t should be no lower than the original basis kernel.

---

> > ### Comment · Reviewer_3dDh · 2023-11-22
> >
> > The response partly addresses my concerns.
> >
> > The effect of RefConv on more recent CNNs is still unclear for me.
> >
> > As the authors only provided the results of FasterNet-S and ConvNeXt-T, at the very least, larger models from the same family should be reported.
> >
> > Additionally, I suggest that the authors consider including more modern CNNs to demonstrate that RefConv is a general method, rather than effective only on some "classic" CNNs.
> >
> > Also, I recommend using [1] as the baseline for ResNet.
> >
> > Given the current rebuttal, I keep my rating unchanged.
> >
> > [1] ResNet Strikes Back: An Improved Training Procedure in Timm.

---

> > > ### Author Response · Authors · 2023-11-22
> > > **Response to updated reviews of Reviewer 3dDh**
> > >
> > > Dear Reviewer 3dDh,
> > >
> > > Thank you for your response. We understand your concerns but unfortunately, the author response phase is nearing its end, and we regretfully acknowledge our inability to promptly finish the experiments with larger models, which require several days for completion. Nevertheless, we are committed to addressing your concerns and will continue to conduct these requested experiments with RepLKNet, SLaK, and bigger ConvNeXt/FasterNet models. **We promise that the results will be added to the revised paper.**
> > >
> > > We kindly request your understanding of our inability to finish the experiments before the author response phase deadline, because we provided a detailed response to your concerns (which were mostly focused on the mechanisms of the proposed method) **on Nov 14th**. Without your timely feedback to our last response, **we did not expect you would request experiments with larger models or TIMM-trained models**.
> > >
> > > While we may not be able to complete such large-scale experiments before the end of the author response phase, we are lucky that Reviewer Z4v5 had also requested additional experiments in the initial review, which we have conducted throughout the rebuttal phase (so that we do not have enough GPU resources for RepLKNet, SLaK, and bigger ConvNeXt/FasterNet models)  and included in the response to Reviewer Z4v5.
> > >
> > > Specifically, we have conducted experiments of ResNet-34 and MobileNetv3 **with the official training scripts of TIMM**. For the well-trained ResNet-34, RefConv improves the accuracy from 75.12% to 75.86%. Similarly, for MobileNetv3, RefConv enhances the accuracy from 74.28% to 74.91%.
> > >
> > > These experiments support the claims made in our paper and **align with your required experiments that use ResNet trained with TIMM as the baseline**. We believe these results should be duly considered for their relevance.
> > >
> > > Thank you for your attention and consideration.
> > >
> > > Please let us know if there is anything we can do to convince you to further raise the score.
> > >
> > > Best Regards,
> > >
> > > Authors

---

> ### Comment · Area_Chair_ztFG · 2023-11-21
>
> Reviewer 3dDh, did the authors' rebuttal had addressed your concerns?
>
> Please reply and post your final decision as well.
>
> AC

---

### Meta-Review · Area_Chair_ztFG · 2023-12-09

**Metareview:**

The PAPER introduces Re-parameterized Refocusing Convolution (RefConv), a new module designed to enhance pre-trained models' performance without additional inference costs, by reorienting convolution kernels to focus on novel representations, thereby improving efficiency and effectiveness in tasks like image classification and object detection. One reviewer recommends acceptance and the other two recommend rejection. The common concern from the reviewers is the effectiveness of the new operator. There are not enough comparisons on stronger baselines with more parameters and training time. After carefully reading the reviews, rebuttal and the paper, the AC agrees with the majority of the reviewers on rejecting the paper.

**Justification For Why Not Higher Score:**

There are concerns about the experiment results to show the effectiveness of the approach.

**Justification For Why Not Lower Score:**

N/A

---

### Decision · Program_Chairs · 2024-01-16

Reject